# Redefining the Incidence and Profile of Fluoropyrimidine-Associated Cardiotoxicity in Cancer Patients: A Systematic Review and Meta-Analysis

**DOI:** 10.3390/ph16040510

**Published:** 2023-03-30

**Authors:** Yajie Lu, Wei Pan, Shizhou Deng, Qiongyi Dou, Xiangxu Wang, Qiang An, Xiaowen Wang, Hongchen Ji, Yue Hei, Yan Chen, Jingyue Yang, Hong-Mei Zhang

**Affiliations:** 1Department of Clinical Oncology, Xijing Hospital, Air Force Medical University, Xi’an 710032, China; 2The State Key Laboratory of Cancer Biology, Biotechnology Center, School of Pharmacy, Air Force Medical University, Xi’an 710032, China; 3The Department of Biomedical Engineering, Air Force Medical University, Xi’an 710032, China

**Keywords:** fluoropyrimidine, cardiotoxicity, adverse event, incidence, meta-analysis

## Abstract

Aim: The cardiac toxicity that occurs during administration of anti-tumor agents has attracted increasing concern. Fluoropyrimidines have been used for more than half a century, but their cardiotoxicity has not been well clarified. In this study, we aimed to assess the incidence and profile of fluoropyrimidine-associated cardiotoxicity (FAC) comprehensively based on literature data. Methods: A systematic literature search was performed using PubMed, Embase, Medline, Web of Science, and Cochrane library databases and clinical trials on studies investigating FAC. The main outcome was a pooled incidence of FAC, and the secondary outcome was specific treatment-related cardiac AEs. Random or fixed effects modeling was used for pooled meta-analyses according to the heterogeneity assessment. PROSPERO registration number: (CRD42021282155). Results: A total of 211 studies involving 63,186 patients were included, covering 31 countries or regions in the world. The pooled incidence of FAC, by meta-analytic, was 5.04% for all grades and 1.5% for grade 3 or higher. A total of 0.29% of patients died due to severe cardiotoxicities. More than 38 cardiac AEs were identified, with cardiac ischemia (2.24%) and arrhythmia (1.85%) being the most frequent. We further performed the subgroup analyses and meta-regression to explore the source of heterogeneity, and compare the cardiotoxicity among different study-level characteristics, finding that the incidence of FAC varied significantly among different publication decades, country/regions, and genders. Patients with esophagus cancer had the highest risk of FAC (10.53%), while breast cancer patients had the lowest (3.66%). The treatment attribute, regimen, and dosage were significantly related to FAC. When compared with chemotherapeutic drugs or targeted agents, such a risk was remarkably increased (χ^2^ = 10.15, *p* < 0.01; χ^2^ = 10.77, *p* < 0.01). The continuous 5-FU infusion for 3–5 consecutive days with a high dosage produced the highest FAC incidence (7.3%) compared with other low-dose administration patterns. Conclusions: Our study provides comprehensive global data on the incidence and profile of FAC. Different cancer types and treatment appear to have varying cardiotoxicities. Combination therapy, high cumulative dose, addition of anthracyclines, and pre-existing heart disease potentially increase the risk of FAC.

## 1. Introduction

Fluoropyrimidines including 5-FU, its oral pro-drug capecitabine, the recent compound preparations S-1 (tegafur), and TAS 102 have been used in the treatment of solid tumors for more than half a century [1,2]. Although numerous therapeutic strategies have been introduced in recent years, such as targeted therapy [3] and immunotherapy [4], fluoropyrimidines are still the fundamental chemotherapeutic agents for treating many tumors [5], playing a significant role in disease control and prolongation of patient survival. However, serious adverse effects (AEs) associated with fluoropyrimidines may result in the modification of the prescribed treatment, or even its interruption. In particular, fluoropyrimidine-associated cardiotoxicity (FAC) is a potential threat for an effective chemotherapy, increasing the mortality for patients predicted to have long-term oncologic survival [6,7]. In recent years, cardiovascular toxicity related to cancer treatment has gained more and more clinical concerns. With the rise of the new discipline “cardio-oncology”, treatment-related cardiotoxicity, such as that of anthracyclines and immune checkpoint inhibitors (ICIs), has become a growing concern in recent decades [8,9]. Cardiotoxicity induced by fluoropyrimidine, however, has not received equal attention (Appendix A). In fact, fluoropyrimidine seems to be one of the most common agents causing cardiotoxicity, second only to anthracycline [10,11]. The reported incidence of FAC ranges from 0 to 68% [12,13,14,15]; however, these data are imprecise and valueless for clinical reference due to the risk of bias. The described cardiac AEs of FAC encompass a broad spectrum, including ischemia, arrhythmia, angina, heart failure, cardiac arrest, enzyme change, cardiomyopathy, myocarditis, and so on [13]. Unfortunately, uncertainty remains regarding the profile of FAC within the published data.

The accurate estimation of the true incidence of FAC and profile of cardiac AEs is essential for preventing severe events and ensuring therapeutic efficacy, and is critical for scientific research. In this study, we represented a meta-analysis across mono or combined fluoropyrimidine therapies, in order to obtain the exact incidence of FAC and profile of cardiac AEs, and to explore further the potential differences in FACs among a variety of demographic characteristics, tumors, study designs, and treatment regimens.

## 2. Materials and Methods

### 2.1. Search Strategy and Selection Criteria

A systematic literature search was performed to identify studies that evaluated fluoropyrimidine-related cardiotoxicity. PubMed, Embase, Medline, Web of Science, Cochrane library, and clinical trials (https://clinicaltrials.gov/ accessed on 20 March 2023) were retrieved from the establishment of each database/website to 31 October 2022, with no language restrictions. The reference lists of the relevant articles were also reviewed to avoid the omission of eligible studies. The retrieval scheme used for each database is shown in Appendix A.

Studies eligible for inclusion met all the following criteria: (1) definitive diagnosis of solid malignances; (2) involving fluoropyrimidine-based treatment, including 5-FU, capecitabine, S-1, and/or TAS 102; (3) the required data were available; (3) treatment-related cardiotoxicity was explicitly reported; (4) prospective or retrospective clinical study. The following exclusion criteria were used: (1) phase I clinical trials; (2) the sample size was smaller than 10; (3) animal experiments or laboratory research; (4) reviews, meta-analyses, comments, or case reports. This work was performed under the guidance of the PRISMA statement [16], and the protocol was registered in PROSPERO (No. CRD42021282155) [17].

### 2.2. Methodological Quality Assessment

The quality assessment tool of the National Institutes of Health (NIH) (Appendix A) was used to assess the methodological quality of the included studies [18]. Each single-arm study was assessed according to a list of 9 items, while the controlled study was assessed against a list of 14 items. For each item, reviewers could select “YES”, “NO”, or “Cannot Determine/Not Applicable/Not Reported”. Based on their responses, each study was then graded as being of “good”, “fair”, or “poor” quality.

### 2.3. Outcomes of Interest and Data Extraction

The main outcome was the incidence of the cardiotoxicities for all grades, grade 3 or higher, and grades 1–2. The profiles of cardiac AEs were also prespecified as important secondary outcomes. The data of basic characteristics, treatment details, and clinical results of cardiotoxicity were obtained from each included study.

### 2.4. Statistical Analysis

The meta-analyses were conducted using R software (Version 4.0.6) with “meta”, “rmeta”, and “metafor” packages. Shapiro–Wilk normality tests on the raw rate and transformed data (log, logit, arcsine, and Freeman–Tukey transformation) were used to determine the most appropriate data type for the pooled analysis. The inter-study heterogeneity was detected by the Cochran’s Q test reporting I^2^ statistic and *p* values. A random-effect model was adopted in case of an indication of significant heterogeneity (I^2^ > 50% or *p* < 0.1) [19,20], otherwise, the fixed-effect model was used. Subgroup analyses were performed based on study-level moderators in order to compare the incidence of FAC among studies with different characteristics. A multilevel meta-regression analysis was conducted to detect the source of heterogeneity further and examine the influence of the moderator variables. Funnel plot and Egger’s tests were used to assess publication bias. The sensitivity analysis was performed by excluding studies one by one to determine the stability of the results of the meta-analysis. 

## 3. Results

### 3.1. Eligible Studies and Characteristics

A total of 211 eligible studies involving 63,186 patients were included [15,21,22,23,24,25,26,27,28,29,30,31,32,33,34,35,36,37,38,39,40,41,42,43,44,45,46,47,48,49,50,51,52,53,54,55,56,57,58,59,60,61,62,63,64,65,66,67,68,69,70,71,72,73,74,75,76,77,78,79,80,81,82,83,84,85,86,87,88,89,90,91,92,93,94,95,96,97,98,99,100,101,102,103,104,105,106,107,108,109,110,111,112,113,114,115,116,117,118,119,120,121,122,123,124,125,126,127,128,129,130,131,132,133,134,135,136,137,138,139,140,141,142,143,144,145,146,147,148,149,150,151,152,153,154,155,156,157,158,159,160,161,162,163,164,165,166,167,168,169,170,171,172,173,174,175,176,177,178,179,180,181,182,183,184,185,186,187,188,189,190,191,192,193,194,195,196,197,198,199,200,201,202,203,204,205,206,207,208,209,210,211,212,213,214,215,216,217,218,219,220,221,222,223,224,225,226,227,228,229,230] (Figure 1). Table 1 summarizes the characteristics of the included studies. The involving population covered 31 countries or regions in the world (Figure 2). According to the NIH quality assessment tools, 61 articles (28.9%) had a good quality score, 150 (71.1%) fair quality, and none was classified as poor (high risk of bias). The detailed information of each included study is shown in Appendix A.

### 3.2. Pooled Incidence of FAC

Our analysis generated robust data on FAC incidence. A total of 186 studies with 40,170 patients were enrolled in the pooled analysis of all-grade cardiac AEs, and 2285 (5.68%) patients experienced at least one cardiac AE. The pooled incidence of all-grade FAC was 5.04% (95% CI 4.21–5.94%) (Appendix A). The pooled incidence of cardiac AEs grade 3 or higher was 1.5% (95% CI 1.09–1.96%), involving 127 studies with 25,273 participants (Appendix A). A total of 718 individuals had cardiac AEs grade 1–2, with a pooled incidence of 2.33% (95% CI 1.57–3.21%) (Appendix A).

### 3.3. Profile of the Cardiac AEs

The profile of FAC includes a variety of disease and symptoms, and, in our study, more than 38 different types of cardiac AEs were reported (Figure 3). From the results of this analysis, cardiac ischemia and arrhythmia were the two most common AEs, occurring in 2.24% (95% CI 1.41–3.00%) and 1.85% (95% CI 1.03–2.62%) of patients, respectively. Heart failure developed in 0.65% of the population (95% CI 0.21–0.82%), probably as a result of severe events following cardiac ischemia and arrhythmia. The left ventricular ejection fraction (LVEF) decreased ≥20% in 1.5% (95% CI 0.6–2.67%) of patients, and 0.91% (95% CI 0.41–1.54%) had cardiac dysfunction (LVEF < 50%). ECG alterations were reported in 37 studies, with a pooled incidence of 5.85% (95% CI 3.4–8.9%), slightly higher than the incidence of FAC (5.18%), indicating asymptomatic ECG alterations in a subset of the population. The ST-T change was the most frequently observed (5.04%, 95% CI 2.66–8.13%), which represents an indicator of cardiac ischemia. Additionally, serum biochemical changes were reported in 16 studies, with a pooled incidence of 1.5% (95% CI 0.69–2.61%).

### 3.4. FAC-Related Deaths

The pooled mortality of FAC was 0.12% (95% CI 0.08–0.15%), involving 114 cases (0.29%, 114/39, 455) from 194 studies. The most frequent causes of cardiotoxicity-related death were sudden cardiac arrest (28.95%) and myocardial infarction (27.19%). Heart failure (15.79%) and severe arrhythmias (14.04%) were also common causes of cardiac death (Table 2).

### 3.5. Factors Influencing the Occurrence of FAC-Subgroup Analysis

#### 3.5.1. Basic Demographic and Study-Level Factors

The results of the subgroup analysis are shown in Figure 4. The incidence of FAC has significantly increased in the past three decades (χ^2^ = 7.8, *p* = 0.02). Studies conducted in Asia outlined a higher incidence than in Europe (χ^2^ = 4.44, *p* = 0.03) and America (χ^2^ = 4.45, *p* = 0.03). No significant difference was observed between their subgroups in terms of study design, trial phase, population age, and methodological quality. Sixty-three studies only included females with breast cancer, producing a lower incidence of FAC than those involving both females and males (χ^2^ = 8.75, *p* < 0.01). Studies that excluded people with pre-existing cardiac disorders had a reduced incidence of FAC (χ^2^ = 4.29, *p* = 0.04).

#### 3.5.2. FAC for Different Cancers

The incidence of FAC varies among different cancers (Figure 4A). The highest pooled incidence was observed in esophagus cancer (10.53%, 95% CI 5.8–16.35%), significantly greater than breast cancer (3.66%, χ^2^ = 8.04, *p* < 0.01) and colorectal cancer (4.59%, χ^2^ = 5.24, *p* = 0.02). The lowest incidence of FAC occurred in breast cancer (3.66%, 95% CI 2.4–5.12%), but no statistical difference was identified when compared with colorectal cancer (χ^2^ = 1.95, *p* = 0.16), head and neck cancer (5.52%, χ^2^ = 1.79, *p* = 0.18), gastric cancer (4.66%, χ^2^ = 0.650, *p* = 0.42), and pancreatic cancer (4.94%, χ^2^ = 0.08, *p* = 0.77). The lung cancer subgroup had the second highest incidence of cardiotoxicity (6.31% 95% CI 2.21–12.06%); however, these studies were from the 1990s, and fluoropyrimidines are now no longer recommended for use in lung cancer.

#### 3.5.3. FAC for Different Treatment Parameters

Significant differences in FAC incidence existed between treatment parameters. Fluoropyrimidines for advanced/metastatic/relapsed diseases had a higher incidence of FAC compared with neoadjuvant or adjuvant treatments (χ^2^ = 6.91, *p* = 0.03); however, no statistical difference was observed between different treatment lines (first-line vs. ≥ second-line, χ^2^ = 0.05, *p* = 0.82). The 5-FU induced cardiotoxicity in monotherapy was significantly lower than that in the combination therapy, either combined with other chemotherapeutic drugs (χ^2^ = 10.15, *p* < 0.01) or targeted agents (χ^2^ = 10.77, *p* < 0.01).

Anthracycline agents and anti-angiogenic drugs are known to cause cardiotoxicity, and our results showed a significantly higher incidence of cardiotoxicity when fluoropyrimidines were combined with anthracyclines (χ^2^ = 4.02, *p* = 0.04) or anti-angiogenic targeted agents (χ^2^ = 15.73, *p* < 0.01) (Figure 4B). The incidence of capecitabine-induced FAC was slightly higher than that of 5-FU (3.44% vs. 2.85%), but without statistical significance (χ^2^ = 0.01, *p* = 0.97). In addition, the compound preparations S1 and TAS 102 showed a relatively low incidence of FAC (S1 2.28%, TAS 102 0.56%).

The occurrence of cardiotoxicity was closely related to drug administration patterns (χ^2^ = 12.29, *p* = 0.03) (Appendix A). The continuous 5-FU infusion for 3–5 consecutive days produced the highest incidence (7.3%). The second highest occurred at the dosage pattern of bolus infusion, followed by continuous infusion (7.09%). Non-consecutive infusion (1.68%), 24 h continuous infusion (3.69%), and continuous infusion on d1,8 or d1 (2.02%) resulted in a relatively low incidence. In fact, a significant positive correlation was identified between the cumulative 5-FU dose per cycle and the cardiotoxicity (χ^2^ = 8.41, *p* = 0.04) (Appendix A). A dosage greater than 3000 mg/m^2^ resulted in a three-fold higher toxicity than the lower dose (≤1000 mg/m^2^) (χ^2^ = 7.66, *p* < 0.01).

### 3.6. The Results of Meta-Regression Analysis

A meta-regression analysis was performed to identify the factors influencing FAC incidence further. The univariable meta-regression of continuous data revealed that female proportion (negative, Q = 8.59, *p* < 0.01) and 5-FU dosage (positive, Q = 9.57, *p* < 0.01) were strongly correlated with FAC (Appendix A). Publication year and median age of population were also correlated with the cardiotoxicity, but the differences were not significant (*p* = 0.058; *p* = 0.071) (Appendix A). Results of the multilevel meta-regression analysis are shown in Table 3. As can be seen in the results, the moderators of publication year (*p* = 0.02), country/region (*p* < 0.01), pre-existing cardiac disorders (*p* < 0.01), study design (*p* = 0.024), treatment attribute (*p* < 0.001), cancer type (*p* = 0.026), regimen (*p* = 0.043), and anthracycline combination (*p* = 0.027) were significant predictors influencing the occurrence of cardiotoxicities. This multilevel meta-regression model totally explained more than half of the inter-study heterogeneity (R^2^ = 51.74%).

### 3.7. Publication Bias and Sensitivity Analysis

No obvious asymmetry was observed in the funnel plots of the main outcomes, suggesting no evidence of significant publication bias, which was confirmed by the Egger’s test (Appendix A). The results of the sensitivity analysis showed that no individual study substantially influenced the pooled results of the above main outcomes (Appendix A), indicating that the results of this meta-analysis were relatively stable.

## 4. Discussion

This study generated robust epidemiological data of FAC incidence and profile based on a single-rate meta-analysis of 211 studies and 63,186 patients, which revised the previous over- or under-estimation of FAC [31,38,231]. We further identified several factors influencing the occurrence of FAC through subgroup analysis and meta-regression analysis. Cardiotoxicity caused by fluoropyrimidines has not been widely recognized and studied in the past. However, along with the wide application of fluoropyrimidine-combined therapy and the increasing demand of long-term mediation for advanced-stage patients with improved survival, increasingly attention has been paid to the cardiotoxicity problem of fluoropyrimidines. The incidence of FAC (5.04%), by our meta-analytic, is second only or even comparable to the familiar cardiotoxicity of anthracyclines (data from a meta-analysis based on 50 thousand cases: The incidence of 6.3%) [232]. Indeed, the real prevalence of FAC might be even higher than what we reported, since not all included studies undertook the most comprehensive cardiac evaluation, which might have resulted in missed FAC detections. Therefore, FAC must be given a high level of attention in clinical practice, as capecitabine, 5-FU, and its modified agents are increasingly used in maintenance therapy for malignancies. The management of FAC in cancer patients has a tremendous impact on the type of antitumor therapies, as well as long-term morbidity and mortality [233].

The profile of fluoropyrimidine-related cardiac AEs has been clearly depicted in our results, which provided evidence for close monitoring and early identification of pertinent cardiac symptoms and signs. Cardiac ischemia is the typical and most common FAC, occurring in 2.24% of patients, and one of the most common causes of fluoropyrimidine-related death (27.19%). The typical ST-T ECG changes were observed in most patients with cardiac ischemia. However, the overall incidence of ECG alterations was even higher than the incidence of cardiac AEs (5.85% vs. 5.04%), indicating that some subjects presented asymptomatic ECG changes, which is often a warning signaling an imminent ischemic event. Several studies considered patients who underwent 24 h ECG Holter monitoring, and, as a result, ischemic ECG changes were observed in 31–68% of patients [233,234]. Therefore, continuous ECG monitoring should be strongly recommended in patients subjected to fluoropyrimidines as the simplest and most useful method for early recognition of FAC. Arrhythmia, including atrial fibrillation, tachycardia, bradycardia, and conduction disorder were also indicated as the common features of FACs (1.85%), which might be a result of cardiac apoptosis triggered by ischemic change and cardiomyocyte damage [235]. LVEF can be used as an indicator of cardiac pump function, which is closely related to heart failure [236]. Our results showed a reduction in LVEF ≥ 20% in 1.5% of patients, while 0.91% developed cardiac dysfunction. However, the evaluation of LVEF using echocardiography is limited by the intrinsic operator-dependency of this technique. Thus, multi-level detection (e.g., ECG, echocardiography, myocardial enzyme, fMRI, coronary angiography, and radionuclide ventriculography) would be helpful for improving the sensitivity required to identify FAC.

Pre-treatment FAC risk assessment should ideally be performed, and the HFA-ICOS risk assessment tools can be considered [233]. Several studies suggested that FACs varied with the population characteristics, cancer types, and treatment regimens [237]. In contrast, other studies found that these factors were insignificant [95]. Our study performed a comprehensive sub-analysis to assess the influences of these variations on FAC, which is useful for indicating the risk factors of FAC. Being female was reported to be a protective factor for myocardial infarction and ischemia [238]. As shown in our results, the incidence of FAC decreased as the proportion of females increased, and the female-only population had a significant lower incidence than the general population. The lower cardiotoxicity was probably due, to some extent, to the protective effect of female hormones. On the other hand, it could also be related to the specific cancer type and regimens, since all patients in the female-only subgroup suffered from breast cancer, and most of them underwent capecitabine-based regimens (1000 mg/m^2^, d1–14, Q3w).

Previous studies reported a higher incidence of FAC in the elderly population [42], and it was reported that patients with an age ≥ 80 years are at great risk of treatment-related cardiac problems [233]. Similarly, we found a positive correlation between median age and cardiotoxicity, but no significant difference was detected in the subgroup analysis. Moreover, only four studies were included in the elderly subgroup (≥60 years) involving only 260 patients. Thus, it was not possible in this work to demonstrate whether the elderly is at particularly risk of FAC. Further large-scale stratified analyses focusing on the elderly are required. Notably, studies performed in Asia showed a higher incidence of cardiotoxicities than those in Europe and America. This result was consistent with the previous findings of Peng et al. [44], in which the incidence of cardiotoxicity in the Chinese population was higher than that in the non-Chinese population (25% vs. 19.9%). Such discrepancies may be derived from the genetic polymorphisms characterizing different ethnicities. For example, the frequency of DPD enzymatic activity varies greatly among Asian, Caucasian, and African-American populations, and it is an indicator of a remarkable risk for cardiac damage [239]. In addition, it is not surprising that pre-existing cardiac disorder significantly increased the risk of cardiotoxicities. One potential reason is that the unrecovered disease causes damage to the cardiomyocytes, making them more sensitive to an external stimulus like 5-FU infusion. Therefore, special care should be taken when administering fluoropyrimidines to patients with underlying cardiac disorders.

Our results indicated that the incidence of FAC differed among tumors. The highest incidence of cardiotoxicities was observed in esophageal cancer (10.53%), whereas the lowest was found in breast cancer (3.66%). Meydan’s study also showed a similar result [124], in which no patient with breast cancer suffered from cardiotoxicity, while the incidence in other tumors, such as colorectal cancer, was more than 3.5%. The discrepancy in cardiotoxicity among tumors may also be derived from gender, treatment regimen, or dosage schedule. For example, the addition of radiotherapy for esophageal cancer greatly increases the treatment-related cardiotoxicity. However, further studies on the FAC among different tumors are required because of the lack of direct comparisons between tumors.

The occurrence of cardiotoxicity varied among different fluoropyrimidine regimens. Combination therapy with other chemotherapy agents or targeted drugs resulted in a higher incidence of FAC than monotherapy. The increased cardiotoxicity could be due to the additional or synergistic deleterious cardiac effects from multiple agents. For example, it is worth noting that the cardiotoxicity significantly increased when fluoropyrimidine was coupled with cardiotoxic drugs such as anthracyclines, which has been approved as a risk for the occurrence of adverse cardiac events [233]. Therefore, when chemotherapy regimens containing anthracyclines are considered, close attention should be paid. In addition, our findings observed that cardiotoxicity induced by capecitabine was not significantly higher than 5-FU, although some previous studies showed a higher toxicity in capecitabine treatment [44,95]. The incidence of FAC differs by 5-FU administration (continuous infusion or bolus infusion), and it occurs in a cumulative dose-dependent manner [147], which was confirmed by our results. In view of the prominent cardiotoxicity of fluoropyrimidines, it is important to develop safe and efficacious new fluoropyrimidine drugs. TAS-102, a novel fluorouracil agent, is a promising safe option for patients. In addition, raltitrexed (Tomudex^®^) can also be used as an alternative to conventional fluorouracil drugs, with a lower incidence of FAC [240].

This study contains some limitations. The first is the heterogeneity at the study level. Although series subgroup analyses and meta-regression were performed to explore the source of heterogeneity further, they did not explain all the heterogeneity of the pooled effects. The clinical heterogeneity of the participants in the included studies could potentially have induced heterogeneity in the results of this meta-analysis. The attributes of the included studies could also have led to unstable results. For example, prospective RCT research can produce more robust conclusions in theory. Secondly, although all the reported cardiac AEs were diagnosed and graded based on the general NCI or WHO criteria, cardiotoxicity might have been underestimated in some studies due to the lack of cardiac-specific examinations. Finally, most of the included studies, especially the clinical trials, adopted strict inclusion criteria, limiting the generalizability of our results for patients outside the inclusion criteria (e.g., patients younger than 18 years or older than 70 years). Hence, real-word studies based on a large-scale population and robust analysis should be necessary to confirm our findings.

## 5. Conclusions

In conclusion, this meta-analysis systematically and comprehensively redefined the incidence and profile of FAC. FAC is not as rare as it seems, and involves a wide variety of related cardiac AEs, with cardiac ischemia and arrhythmia being the most common. The occurrence of cardiotoxicity varies among different publication decades, country/regions, genders, cancer types, and treatment details. Combination therapy, high cumulative dose, addition of anthracyclines, and pre-existing heart disease potentially increase the risk of FAC. This global overview of cardiotoxicity in fluoropyrimidine-based treatment may be used as a clinical reference in clinical practice for the management of cancer therapy.

## Figures and Tables

**Figure 1 pharmaceuticals-16-00510-f001:**
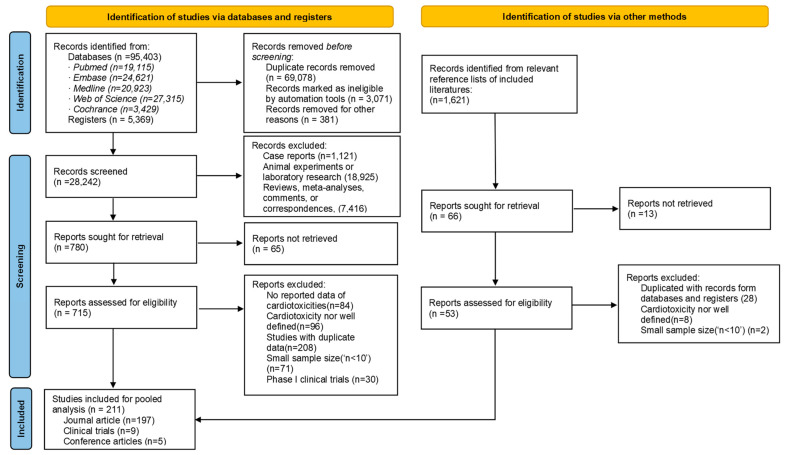
Flow diagram of the inclusion/exclusion process of the relevant literature with number of articles at each step.

**Figure 2 pharmaceuticals-16-00510-f002:**
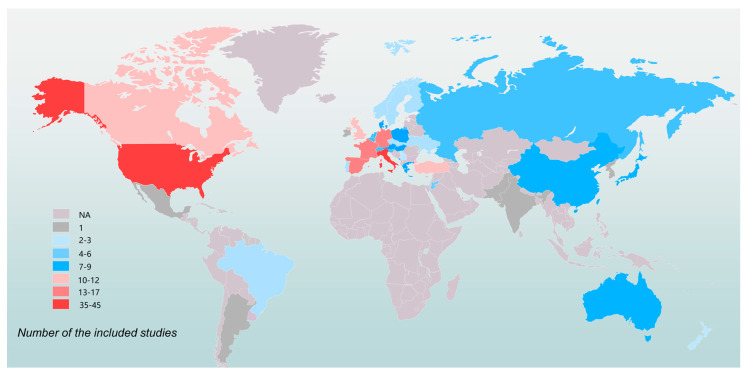
Geographical distribution of the 211 included studies.

**Figure 3 pharmaceuticals-16-00510-f003:**
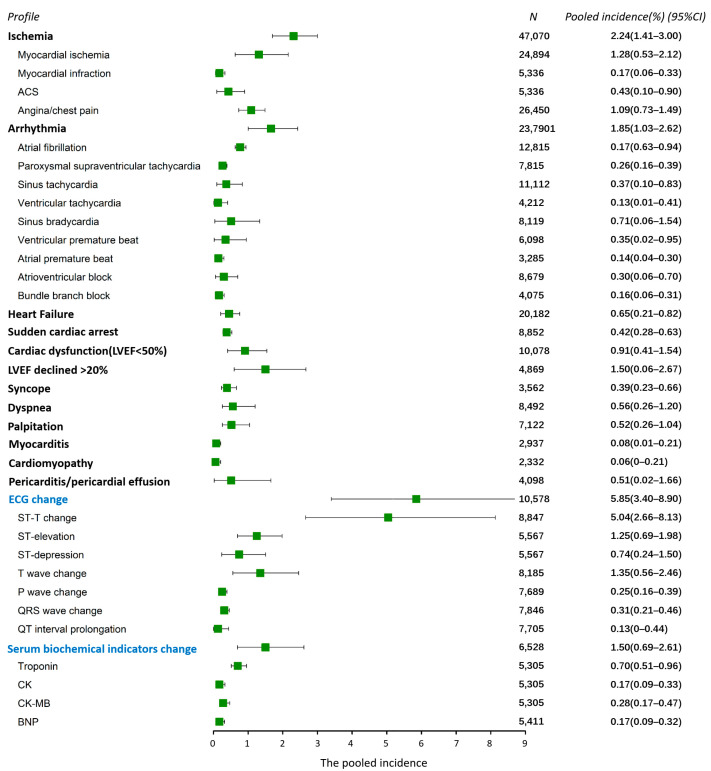
The profile of the FAC (all-grade cardiac AEs). Each bar and square represent the pooled value of incidence and 95% confidence interval for one cardiac AE. Vertical line indicates the overall incidence of all-grade adverse events (5.04%). Abbreviations: ACS, acute coronary syndrome; LVEF, left ventricular ejection fraction; CK, creatine kinase; BNP, B-type natriuretic peptide. “*N*” at the top of this figure represents the number of patients included in the evaluation for each AE.

**Figure 4 pharmaceuticals-16-00510-f004:**
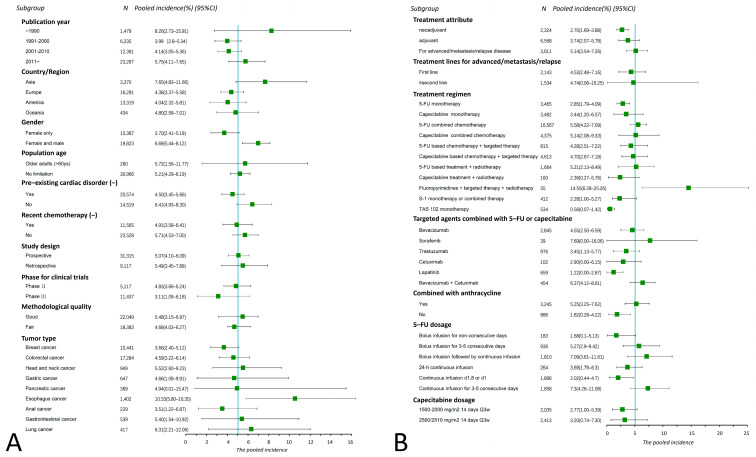
The results of subgroup analysis outcomes of all-grade FAC by basic characteristics (**A**) and treatment details (**B**) of included studies. Each bar and square represent the pooled value of incidence and 95% confidence interval for one subgroup. Vertical line indicates the pooled incidence of all-grade FAC (5.04%); “*N*” at the top of this figure represents the number of patients included in the evaluation for each subgroup.

**Table 1 pharmaceuticals-16-00510-t001:** Characteristics of the 206 studies included in this meta-analysis.

	No. of Studies (*n*)
**Year Published**	
Before 1980	1
1981 to 1990	7
1991 to 2000	63
2001 to 2010	77
2011 to 2020	57
After 2020	6
**Study Design**	
Retrospective	34
Prospective ^a^	177
Phase ii trial	81
Phase iii trial	34
**Population Age**	
No limitation	185
Older adults (>60 years)	6
NA	20
Gender	
Female only	67
Female and male	120
NA	24
**Tumor Type**	
Breast cancer	66
Colorectal cancer	56
Gastric cancer	15
Esophagus cancer	11
Head and neck cancer	11
Gastrointestinal cancer	5
Pancreatic cancer	4
Lung cancer ^b^	4
Anal cancer	4
Solid malignancies ^c^	28
Others ^d^	7
**Treatment Regimen ^e^**	
5-FU monotherapy	23
5-FU combined chemotherapy	114
5-FU based chemotherapy + targeted therapy	15
Capecitabine monotherapy	20
Capecitabine combined chemotherapy	23
Capecitabine based chemotherapy + targeted therapy	28
S-1 monotherapy or combined therapy	5
TAS 102 monotherapy	1
Fluoropyrimidines + radiotherapy	25
Fluoropyrimidines + immunotherapy	1
Mixed fluoropyrimidines ^f^	8
Fluoropyrimidines + targeted therapy + radiotherapy	1
**Treatment Attribute**	
Neoadjuvant	24
Adjuvant	27
For advanced/metastatic/or relapsed disease	58
First line	32
≥second line	8
**Methodological Quality ^g^**	
Good	61
Fair	150
Poor	0

Notes: NA, not available; ^a^, there were 177 prospective studies, of which 115 were clearly marked as clinical trials (81 phase II trials and 34 phase III trials), and the remaining 62 were not indicated as clinical trials; ^b^, fluoropyrimidine is no longer recommended for use in lung cancer, and the included four studies were all in the 1990s; ^c^, solid malignancies: including two or more tumor types, such as breast cancer, colorectal cancer, gastric cancer, head and neck cancer, and so on; ^d^, others: including renal cell carcinoma (*n* = 1), prostate cancer (*n* = 1), ovarian cancer (*n* = 1), hepatocellular carcinoma (*n* = 1), bladder cancer (*n* = 1), biliary tract cancer (*n* = 1), and adrenal cortical carcinoma (*n* = 1); ^e^, the data were calculated based on 265 arms from 211 included studies; ^f^, mixed fluoropyrimidines: does not specifically refer to one single drug, containing two or more fluorouracil drugs; ^g^, the methodological quality was assessed by the quality assessment tool of the National Institutes of Health (NIH).

**Table 2 pharmaceuticals-16-00510-t002:** Characteristics of the 114 FAC-related deaths.

Cause of Death	N	(%)
Sudden cardiac arrest	33	28.95%
Myocardial infarction	31	27.19%
Heart failure	18	15.79%
Arrhythmias	16	14.04%
Complete atrioventricular block	4	3.51%
Ventricular fibrillation	1	0.88%
Arrhythmia (specific type NA)	11	9.65%
Pericarditis/pericardial effusion	2	1.75%
Cardiomyopathy	1	0.88%
NA *	13	11.4%

Notes: NA, not available; * Thirteen studies only reported deaths due to cardiotoxicity, but did not provide details about it.

**Table 3 pharmaceuticals-16-00510-t003:** The results of multilevel meta-regression analysis.

Factors	Variables	β	Z Value	*p* Value	
Intrcpt	/	−6.648	−2.432	0.015	
Publication year	Continuous data	0.003	2.326	0.020	*
Age	All range	/	/	/	
	Elderly (≥60 years)	−0.013	−0.196	0.844	
Country/region	America	/	/	/	
	Asia	0.094	2.796	0.005	**
	Europe	0.001	0.023	0.982	
	Oceania	−0.072	−1.161	0.246	
	Multi: Europe/America/Oceania	0.052	1.302	0.193	
Gender	Female and male	/	/	/	
	Female only	−0.031	−0.546	0.585	
Pre-existing cardiac disorder (-) ^a^	Yes	/	/	/	
	No	0.060	2.865	0.004	**
Recent chemotherapy (-) ^b^	Yes	/	/	/	
	No	0.010	0.488	0.625	
Methodological quality ^c^	Fair	/	/	/	
	Good	−0.006	−0.246	0.806	
Study design	Retrospective	/	/	/	
	Prospective	0.063	2.258	0.024	*
Phase for trials	Phase ii	/	/	/	
	Phase iii	−0.026	−0.844	0.399	
Treatment attribute	Neoadjuvant	/	/	/	
	Adjuvant	0.098	2.5112	0.012	*
	For advanced/metastasis/relapse	0.162	3.768	0.000	***
Treatment line ^d^	First line	/	/	/	
	≥second line	0.023	0.368	0.713	
Tumor type	Esophagus cancer	/	/	/	
	Breast cancer	−0.079	−1.055	0.291	
	Colorectal cancer	−0.082	−1.703	0.089	-
	Gastric cancer	−0.142	−2.222	0.026	*
	Head and neck cancer	−0.082	−1.595	0.111	
	Others ^e^	−0.044	−0.854	0.393	
Regimen	5-FU mono	/	/	/	
	Capecitabine mono	0.060	0.972	0.331	
	5-FU-combined chemo	0.069	1.920	0.055	-
	Capecitabine-combined chemo	0.105	1.908	0.056	-
	5-FU-based chemo/target	0.149	2.021	0.043	*
	Capecitabine-based chemo/target	0.089	1.480	0.139	
	5-FU-based chemo/radiotherapy	−0.001	−0.014	0.989	
	Capecitabine-based chemo/radiotherapy	0.030	0.405	0.686	
Targeted agent ^f^	Lapatinib	/	/	/	
	Trastuzumab	0.049	0.742	0.458	
	Bevacizumab	0.016	0.225	0.822	
	Cetuximab	0.078	0.576	0.565	
	Sorafenib	0.017	0.115	0.908	
	Bevacizumab + cetuximab	0.035	0.385	0.700	
5-FU dosage	Bolus infusion for non-consecutive days	/	/	/	
	Bolus infusion for 3–5 consecutive days	0.088	0.825	0.409	
	Bolus infusion followed by continuous infusion	0.069	0.638	0.523	
	Continuous infusion d1,8 or d1	−0.033	−0.299	0.765	
	24 h continuous infusion	0.011	0.090	0.928	
	Continuous infusion for 3–5 consecutive days	0.114	1.059	0.290	
Capecitabine dosage	1500–2000 mg/m^2^ 14 days Q3w	/	/	/	
	2500/2510 mg/m^2^ 14 days Q3w	0.086	1.684	0.092	-
Combined with anthracycline ^g^	No	/	/	/	
	Yes	0.116	2.213	0.027	*

Notes: - *p* < 0.1, * *p* < 0.05, ** *p* < 0.01, *** *p* < 0.001; NA, not available; ^a^, pre-existing cardiac disorder (-): were patients with cardiotoxicity excluded or not? ^b^, recent chemotherapy (-): were patients with recent chemotherapy excluded or not? ^c^, the methodological quality was assessed by the quality assessment tool of the National Institutes of Health (NIH); ^d^, the treatment lines were only for metastasis/advanced/relapsed disease; ^e^, “others” including gastrointestinal cancer, pancreatic cancer, lung cancer, and anal cancer. ^f^, targeted agents combination therapy in breast cancer and colorectal cancer; ^g^, combination therapy with anthracycline in breast cancer.

## Data Availability

The data presented in this study are available upon request from the corresponding author.

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
