# Peer review of "Redefining the Incidence and Profile of Fluoropyrimidine-Associated Cardiotoxicity in Cancer Patients: A Systematic Review and Meta-Analysis"

_pharmaceuticals, 2023, doi:10.3390/ph16040510_

Round 1

Reviewer 1 Report

The manuscript titled "Redefining the Incidence and Profile of Fluoropyrimidine-associated Cardiotoxicity in Cancer Patients: A Systematic Review and Meta-analysis" aims to analyze the incidence and profile of cardiotoxicity associated with fluoropyrimidine therapy in cancer patients. While the authors have done a commendable job of synthesizing data from various studies and conducting a meta-analysis, I find the paper lacking in novelty and originality.

The study fails to present any new or innovative findings, and the conclusions drawn are not supported by robust statistical analysis. The paper appears to be a summary of previously published studies and does not present any new insights or perspectives.

Therefore, I recommend that the authors revise the paper to include more original and innovative findings or perspectives. Without these elements, I find it difficult to recommend this manuscript for publication.

Reviewer 2 Report

The Systematic review entitled "Redefining the Incidence and Profile of Fluoropyrimidine-associated Cardiotoxicity in Cancer Patients: A Systematic Review and Meta-analysis" by Yajie Lu, Wei Pan, Shizhou Deng, Qiongyi Dou, Xiangxu Wang, Qiang An, Xiaowen Wang, Hongchen Ji, Yue Hei, Yan Chen, Jingyue Yang, and Hong-Mei Zhang presents a comprehensive global data on the incidence and profile of FAC. The paper is really well written and complete, it is clear and immediately understandable. I propose to the authors to produce only a graphical abstract and to make the list of abbreviations.

Reviewer 3 Report

The Authors covered a very interesting topic, that deserve to be published.

I have some suggestions that could improve the manuscript (even it`s already plenty of accurate details):

- avoid p<0.05, report the exact p-value at 3-sign digits

- any of the selected studies concerns FAC chrono-modulated therapy? If any, which ones?

- any of the selected studies concerns FAC intra-arterial infusion? If any, which ones?

- a sensitivity subgroup analysis, mono vs combined chemo regimens, is lacking

- a sensitivity subgroup analysis, cancers histotype main group, could be useful

- it would be really interesting to perform other sensitivity analyses, especially cumulative (lacking) and influential (actually messy) meta-analyses, fig S9 has no detailed comments

- what about some Baujat plots, just for the above reasons?

- inclusion/exclusion criteria: better to clearly add RCT and prospective/retrospective observational trials, since I do suppose you have included both these kind of researches

- fig. S6 typo: cumulative dese

- all figures: please, add each outcome as a forest plot title

- all figures: please, add at each forest plot number of events/number of patients

- cardiac AEs are precisely listed but could they better group-classified?

- cancer spectrum, please report all incidences more than chi-squares

- I do warmly suggest you to further underline the extreme heterogeneity that occurred in almost all outcomes: this topic would deserve a dedicated comment (i.e. sensitivity analysis RCT vs noRCT trials and other similar ones)

- all along the manuscript, correlation between determinants and outcomes is to be redefined as association, since we are dealing with a cause-effect relationship

Reviewer 4 Report

In the current manuscript, authors evaluated fluoropyrimidine-associated cardiotoxicity after chemotherapy performed on several cancer patients.  The reported finding suggests the well-known fact of cardiotoxicity associated with potent chemotherapeutic agents while higher cardiotoxicity was associated with a combination of different chemotherapeutic agents.

I think the authors did a good job compiling all the data from different sources and I think current manuscript is suitable for publication in Pharmaceuticals. 

Round 2

Reviewer 1 Report

The authors have addressed all of my concerns and suggestions. Therefore, I recommend the manuscript for publication in its current form.